# 1 High-resolution inventory and classification of retrogressive thaw

# 2 slumps in West Siberia

- Nina Nesterova<sup>1,2</sup>, Ilia Tarasevich<sup>3,4</sup>, Marina Leibman<sup>3</sup>, Artem Khomutov<sup>3</sup>, Alexander Kizyakov<sup>4</sup>, Ingmar
- 4 Nitze<sup>1</sup>, Guido Grosse<sup>1,2</sup>
- <sup>1</sup>Permafrost Research Section, Alfred Wegener Institute, Helmholtz Centre for Polar and Marine Research, Potsdam, 14473,
- 6 Germany
- <sup>2</sup>Institute of Geosciences, University of Potsdam, Potsdam, 14476, Germany
- 8 Earth Cryosphere Institute, Tyumen Scientific Centre SB RAS, 625026, Tyumen, Russia
- <sup>4</sup>Cryolithology and Glaciology Department, Faculty of Geography, Lomonosov Moscow State University, Moscow, 119991,
- 10 Russia

11

29

- 12 Correspondence to: Nina Nesterova (nina.nesterova@awi.de)
- Abstract. Permafrost thaw disrupts ecosystems, hydrology, and biogeochemical cycles, reinforcing climate change through a positive permafrost-carbon feedback loop. Thaw can be gradual, deepening the active layer, or abrupt, triggering thermokarst,
- thermo-erosion, or thermodenudation. Retrogressive thaw slumps (RTSs) are a key manifestation of abrupt permafrost thaw.
- 16 Yet, their distribution, scale, and environmental controls in the West Siberian Arctic remain poorly understood, further
- 17 complicated by their rapid evolution. This study presents an extensive update of the West Siberian RTS inventory through
- manual mapping using high-resolution, multi-source, multi-year recent (2016-2023) satellite basemaps (ESRI, Google Earth,
- 19 and Yandex Maps). We developed an RTS classification capturing key environmental parameters, including morphology,
- spatial organization, terrain position, and associated relief-forming concurrent processes. The dataset comprises 6168 classified
- 21 RTS landforms, integrating newly mapped sites with previously reported occurrences to provide a comprehensive view of a
- 22 445226 km² region covering the Yamal, Gydan, and Tazovsky peninsulas. The collected data underwent manual filtering and
- verification, leveraging local field experience and observations from key sites to reduce uncertainty and minimize false
- 24 positives. Accuracy analysis, performed by comparing the dataset with various field datasets collected across the peninsulas,
- confirmed high accuracy (>90%) for RTS identification. The dataset likely underestimated the distribution of small RTSs due
- to the resolution limitations of remote sensing data, hence generally providing a conservative estimate. This dataset serves as
- a valuable resource for diverse research fields, including ecology, biogeochemistry, geomorphology, climatology, permafrost
- science, and natural hazard assessment. Additionally, it provides a crucial reference dataset for machine learning applications,
  - enhancing upcoming remote sensing classification and predictive modeling approaches.

## 1 Introduction

60

Babkina et al., 2019).

Permafrost is any ground that stays below 0°C for two or more consecutive years (Harris et al., 1988). It constitutes about 15% 32 of the Northern Hemisphere landmass (Obu et al., 2021) and is experiencing significant warming and reduction in extent due 33 to global warming (AMAP, 2017; Biskaborn et al., 2019; Smith et al., 2022). Permafrost thaw not only affects the high-latitude 34 northern ecosystems and hydrological cycle but also releases carbon into the atmosphere and hydrosphere, contributing to 35 global climate change with a positive feedback loop (Schuur et al., 2015). However, permafrost carbon emissions are still 36 poorly integrated into global climate models (Miner et al., 2022). Furthermore, permafrost degradation manifests itself both 37 gradually and abruptly. Gradual thaw slowly deepens the active layer over time (Brown et al., 2000; Luo et al., 2016; Vasiliev 38 et al., 2020), while abrupt thaw in ice-rich permafrost triggers rapid thermokarst or thermo-erosion processes, leading to the 39 formation of various landforms. Prime examples of such abrupt thaw events are specific types of permafrost-region landslides 40 termed retrogressive thaw slumps (RTSs) (Nesterova et al., 2024). 41 RTSs are slope failures formed due to the thaw of exposed ice-rich permafrost (Fig. 1) (Mackay, 1966). These dynamic features 42 can develop in a polycyclic fashion (Lantuit et al., 2005). Usually, the initial stages involve active ice ablation and downslope 43 mudflows, followed by a stage of stabilization and colonization with pioneer vegetation (Mackay, 1966; Kerfoot, 1969; 44 Leibman and Kizyakov, 2007). Active RTS can be considered as one of the clear indicators of permafrost response to increased 45 air temperatures and higher summer precipitation (Lantz and Kokelj, 2008; Kokelj et al., 2015; Leibman et al., 2021; Barth et 46 al., 2025). RTS occurrence significantly impacts the environment by altering the vegetation, topography, hydrology, as well 47 as carbon fluxes (Lantz et al., 2009; Thienpoint et al., 2013; Cassidy et al., 2017). The prediction of RTS occurrence and 48 activity is challenging due to heterogeneous ground ice distribution (Pollard and French, 1980; Makopoulou et al., 2024) across 49 the Arctic, limited observational field data (Ward Jones et al., 2019), and the lack of models capable of simulating RTS 50 initiation and dynamics (Yang et al., 2025). 51 The north of West Siberian Arctic, with its predominantly continuous permafrost distribution (Obu et al., 2019), is 52 characterized by a high abundance of RTS. The prevalence of massive ground ice (Baulin et al., 1967; Streletskaya et al., 2013; 53 Leibman and Kizyakov, 2007; Badu, 2015) that often occurs close to the surface contributes to the widespread abundance of 54 RTSs (Khomutov et al., 2017). Moreover, the observed amplification of seasonal thawing and growth of permafrost 55 temperatures (Babkina et al., 2019; Biskaborn et al., 2019; Vasiliev et al., 2020) presents an additional factor for the mass 56 initiation of RTS in the region. So far, the majority of RTS studies in the north of West Siberia have only been based on 57 fieldwork at local key sites (Leibman and Kizyakov, 2007; Leibman et al., 2015; Khomutov et al., 2017; Novikova et al., 2018; 58 Streletskaya et al., 2018; Babkina et al., 2019). Long-term field observations at the research station "Vaskiny Dachi" in Central 59 Yamal reported the activation of rapid thaw processes after the extreme summer warmth of 2012 (Khomutov et al., 2017;

2

Figure 1 RTS in Central Yamal, West Siberia, Russia. Photo taken in August 2021 by Nina Nesterova.

The vast majority of novel large-scale RTS studies utilize automated mapping with remote sensing data. This automated approach has some limitations for West Siberia so far, including using only a moderate spatial resolution of 30m not sufficient for detecting smaller RTS, only a partial cover of the West Siberian Arctic, the lack of high-resolution ground truth data, a large amount of false positive detection, and further feature interpretation ambiguities (Nitze et al., 2018; Runge et al., 2022; Nitze et al., 2024). Furthermore, the polycyclicity of RTS development results in highly complex spatial patterns characterized by multiple overlapping or nested RTSs (Nesterova et al., 2024), which introduces further difficulties in highly automated mapping efforts. New cutting-edge panarctic datasets building on automated detection methods are being released (DARTS, Nitze et al., 2024b) but still have some limitations in accuracy on the local to regional scale.

In contrast, manual mapping of RTSs with high-resolution imagery by experts with regional knowledge can provide higher accuracy and decrease the amount of false positive detections (Lewkowicz and Way, 2019; Ward Jones et al., 2019; Nitze et al., 2024). A first manually mapped inventory of RTSs in the West Siberian Arctic was performed using the Yandex Maps high to moderate resolution satellite basemap representing the 2016-2018 period (Nesterova et al., 2021). The dataset reports 439 RTSs over both the Yamal and Gydan peninsulas. Due to the different spatial resolutions of satellite images used in the basemap (ranging from 0.4 to 15 m), the results tend to underestimate modern RTS distribution, particularly in areas where

77 only lower resolution imagery was available. Therefore, there was still no full understanding of the scale of thaw slumping in 78 the West Siberian Arctic, its distribution, and environmental parameters, which are further complicated by the rapid evolution 79 of RTSs. 80 We provide an extensive update of the West Siberian RTS inventory for 2021, which was performed by manually mapping 81 RTS in the north of West Siberia using multi-source and multi-year satellite basemaps (high-resolution ESRI, Google Earth, 82 and Yandex Maps satellite basemaps). We further added all the RTS locations reported for this region in the literature so far. 83 The collected dataset was manually filtered and compared to field data. This multi-source approach, in combination with 84 regional field experience and field observations, gathered earlier at various key sites, helped us to minimize the uncertainty 85 and decrease the number of false positive detections. We additionally developed a classification to describe each RTS, capturing their main environmental parameters such as morphology, spatial organization, terrain position, and concurrent 86

# 2 Methodology

relief-forming processes.

Our approach includes four main steps: (1) visual identification of RTS and manual RTS point collection, (2) classification and parameter attribution, (3) iterative correction loop, and (4) final accuracy assessment (Fig. 2). Manual RTS point collection, classification, and correction were performed in QGIS software version 3.14. Accuracy analysis, plotting, and statistical calculations were performed using Python version 3.12.7. Chord diagrams were plotted in R, using RStudio 2024.12.0+467. The resulting points were analysed for clustering using Ripley's K function. Ripley's K function determines whether spatial points have a random, dispersed, or cluster distribution over a certain distance or scale (Dixon, 2001).

Figure 2 Workflow overview. Rectangles with rounded corners present the datasets, and rectangles with sharp corners present the curation steps. The four main stages are numbered: 1 - Visual RTS identification and manual point collection stage, 2 - RTS point classification and parameter attribution, 3 - Iterative correction, and 4 - Accuracy assessment.

## 2.1 RTS point mapping

The study area in the north of West Siberia is 445 226 km² and includes the Yamal, Gydan, and Tazovsky peninsulas (Fig. 3). To ensure the completeness of the RTS dataset we reviewed previously published RTS datasets for the region, all of which were mapped using automated methods (Fig. 3). We manually filtered RTS datasets from Nitze et al. (2018), Runge et al. (2022), Bernhard et al. (2022), Huang et al. (2023), and Nitze et al. (2025) to verify the presence of RTS and ensure that only true positives were included. This verification was conducted using the same available datasets that we later used for manual point collection, as described further below.

Manually collected RTS dataset published in 2021 (Nesterova et al., 2021) was also integrated: points were revised, classified, and renamed.

Figure 3 Study area in West Siberia with RTS datasets previously published in the literature. Please note that none of the external datasets fully covers the entire study area.

For our visual identification and manual collection of RTS points, we created a regular grid of 3.9 \* 3.9 km cells covering the entire study area (Fig. 4a). This cell size was chosen as the optimal for visual inspection of the area and progress tracking, balancing detail and generalization. The ESRI satellite basemap was used as the primary source of information for RTS point collection due to the best quality of its recent very high-resolution imagery. This included high-resolution imagery (up to 0.31 m) largely with low cloudiness and an almost complete absence of visual artifacts. In rare cases, when the ESRI basemap did not fulfill visual quality criteria, such as no clouds, summer time of the image acquisition, and no artifacts, we used the Yandex Maps satellite basemap instead. In exceptional cases when neither the ESRI nor the Yandex basemaps fulfilled the visual quality criteria, we additionally checked the Google satellite basemap.

Figure 4 Manual mapping of RTSs in West Siberia: (a) Example of a grid cell with manually mapped RTSs (orange dots); (b) Coverage of the study area by high-resolution satellite images from different years in the ESRI basemap in km²; (c) Example of a lake shore RTS (marked by yellow point) on ESRI basemap imagery and typical visual RTS indicators: 1 – headwall, 2 – mudflow, 3 - contrasting colors of the disturbed slump floor with bare ground and the surrounding intact tundra vegetation; (d) Example of coastal RTS (marked by yellow point) on ESRI basemap imagery affected by coastal thermo-erosion, with white bracket indicating the full elongated extent of the coastal landform considered to be a single RTS in our inventory dataset. ESRI basemap used in (a), (c), and (d) has the following credits: Esri, DigitalGlobe, GeoEye, i-cubed, USDA FSA, USGS, AEX, Getmapping, Aerogrid, IGN, IGP, swisstopo, and the GIS User Community.

The majority of the high-resolution satellite images used in the ESRI basemap mosaic are recent Maxar images obtained after

2015 (Fig. 4b). Over a third of the study area is covered by satellite images from 2023 (Fig. 4b). Since the ESRI basemap was utilized as the primary source, all metadata related to the satellite images (image date acquisition, image resolution, image accuracy, min and max map level, satellite description, ESRI release name) in the mosaic for identifying the RTS is stored within the inventory dataset's metadata. Yandex Maps basemap presents a mosaic of various satellite imagery taken in 2016-2018 with spatial resolutions ranging from 0.4 up to 15 m. The majority of images are dated July 2017 (Nesterova et al., 2021). For the Google satellite image layer, no individual image metadata was provided.

RTSs were identified at a 1:1000 mapping scale in the satellite imagery based on visual indicators such as a clear outline of the headwall, the presence of a mudflow, and the sharp contrast in colors between the disturbed slump floor with bare ground and the adjacent intact tundra vegetation (Fig.4c). Thus, stabilized RTSs were also identified when the indicators were still visible. For each identified feature, we created a point in the location of the RTS within the visible outlines of the RTS with

the best possible approximation to the visual center of the landform.

Each digitized point represented one feature that would be classified (see Sect. 2.2). Due to the complex nature of coastal RTSs sometimes stretching along coastal segments (Fig. 4d), we decided to identify each elongated contour with visible semicircles embedded inland as one feature. Such contours were often separated from each other by little streams or watercourses. This approach allowed us to utilize a single technique for all coastal RTSs, regardless of their size and shape.

The RTS points underwent two visual corrections by the first author. To differentiate the process of coastal erosion from thermodenudation (Günther et al., 2012; Nesterova et al., 2024) and thereby distinguish other coastal landforms from RTSs, a special correction was applied to all coastal RTSs and thermoterrace RTSs (see Sect. 2.2). This involved verifying the headwall retreat of the RTS outline using the ESRI Wayback Machine - a digital archive of the World Imagery basemap of different versions providing multi-temporal imagery (ESRI Wayback Imagery, 2024). The same verification procedure was applied for the identification of RTS in the southernmost part of our West Siberian study area, where no reliable data on massive ground ice distribution is available and thus permafrost landforms can have different origins. The literature specifies the limits of massive ground ice extent in the north of West Siberia only very approximately (Baulin and Danilova, 1998).

## 2.2 Classification

We classified each RTS point based on terrain position, morphology, spatial organization, and concurrent cryogenic processes (Fig. 5). The four main criteria had a total of 15 parameters.

Figure 5 RTS classification scheme with four main criteria (shown as grey blocks) and 15 variables.

The terrain position of an RTS is defined based on the location of the object to either some hydrological feature (*sea coast*, *river bank*, *lakeshore*, and *gully*) or just *slope* when there was no visible hydrological feature. The location *lake* was selected for RTSs even on the former shores of drained lakes.

We further defined three types of RTS morphologies: *thermocirque*, *thermoterrace*, or a *combination* of these two (Nesterova et al., 2024). *Thermocirque* generally presents a horseshoe-like RTS shape (Fig. 6a), while *thermoterrace* is applied to an elongated RTS with mostly straight headwall outlines parallel to a coastline or riverbank (Fig. 6b). The combination of these two morphologies sometimes occurs when the elongated RTS landform also contains circular isometric curves of headwall outlines (Fig. 6c). It is usually formed when a thermocirque merges with a thermoterrace or when multiple thermocirques

merge in one elongated landform. The complicated shapes of these combined RTS features make it highly challenging to distinguish between individual elongated and horseshoe-like RTSs (Fig. 6c). Our decision tree to define the morphology of RTS is shown in Supplement.

Figure 6 Examples of the three main RTS morphologies mapped in West Siberia: (a) Two thermocirques (yellow dots); (b) A single large thermoterrace (yellow dot); (c) A combined RTS morphology of merged thermocirques or merged thermocirque and a thermoterrace (yellow dot). ESRI basemap used has the following credits: Esri, DigitalGlobe, GeoEye, i-cubed, USDA FSA, USGS, AEX, Getmapping, Aerogrid, IGN, IGP, swisstopo, and the GIS User Community.

Due to the polycyclic nature of RTS development, these landforms can exhibit a very complex spatial organization of nested and amalgamated RTSs (Nesterova et al., 2024). We identified two types of RTS spatial organization: single landforms and complex landforms. RTS can be classified as a single landform when its outline is distinct and clearly defined and there is no more than one actively thawing zone within this outline (Fig. 7a). RTS can be classified as a complex landform when its boundary is difficult to define and/or there are two or more actively thawing zones (Fig. 7b). All the RTSs with combined morphologies were marked as complex landforms.

The influence of concurrent (happening in parallel to RTS development) processes on RTS development is described in Nesterova et al. (2024). For each mapped RTS, we noted the possible presence of 5 concurrent processes: *lateral thermo-erosion, coastal thermo-erosion, ice wedge erosion, nivation,* and *thermokarst subsidence. Lateral thermo-erosion* was identified by the rugged outline of the RTS and visible traces of erosive channels (Fig. 8a). The *Coastal thermo-erosion* classifier includes not only the sea coast erosion but also river and lakeshore erosion. It was determined by a sharp dark outline of the RTS base along the coastline of a waterbody and the absence of sediment accumulation in the water (Fig. 8a). We have noted *ice wedge erosion* when an RTS headwall had a jagged outline resembling the adjacent polygonal surface of undisturbed tundra (Fig. 8b). *Nivation* in the context of this study is considered as persistent snow cover. It was detected as white patches of snowpacks that stayed over the summer within RTS (Fig. 8a). *Thermokarst subsidence* appears as small thermokarst ponds filled with water. It is noticeable as black patches within the RTS outline (Fig. 8b).

Figure 7 Examples of the different spatial organization of RTSs: (a) Single RTS landform with a distinct outline (yellow dot); (b) Complex RTS landform (yellow dot) with multiple nested active (1) and stabilized (2) RTSs within one contour. ESRI basemap used has the following credits: Esri, DigitalGlobe, GeoEye, i-cubed, USDA FSA, USGS, AEX, Getmapping, Aerogrid, IGN, IGP, swisstopo, and the GIS User Community.

Figure 8 Examples of RTS with concurrent processes: (a) Stabilized RTS (yellow dot) at a riverbank. The white arrow (1) points to the clear dark boundary between the RTS and the waterbody, which together with the absence of sediment accumulation, indicates ongoing coastal thermo-erosion at the slump base. The purple arrows (2) point to the rugged outline of RTS and traces of erosive channels, indicating lateral thermo-erosion. The green arrow (3) points at the white patch of the remaining snowpack (nivation). (b) Stabilized RTS (yellow dot) at a lakeshore. The light blue arrows (4) point to the polygonal surface around the RTS and (5) the jagged outline of the headwall suggesting ice-wedge degradation. The orange arrows (6) point to the small black patches of thermokarst ponds within the RTS. ESRI basemap used has the following credits: Esri, DigitalGlobe, GeoEye, i-cubed, USDA FSA, USGS, AEX, Getmapping, Aerogrid, IGN, IGP, swisstopo, and the GIS User Community.

# 2.3 Accuracy assessment

225

#### 2.3.RTS location accuracy

positives and false negatives.

We compared the RTS point locations of our dataset with two sets of ground truth field data to estimate the accuracy of our 207 mapped RTS point locations. 208 The first set of RTS locations was collected for the Vaskiny Dachi Research Station in Central Yamal by Khomutov et al. 209 (2024) and included 158 points. The authors used satellite images of QuickBird-2 for 2010, GeoEye-1 and WorldView-2 for 210 2013, and WorldView-2, 3 for 2018, as well as the results of long-term field observation to map RTSs. Since the RTS mapping 211 protocols can significantly affect the results (Nitze et al., 2024), we have adjusted these ground truth points to align with our 212 mapping protocol in which one point stands for one RTS landform. When comparing our points to the ground truth collected, 213 we observed inconsistencies in mapping RTS points. For example, while the ground truth dataset might contain two or three 214 points for an RTS landform, our approach would place only one. To account for these differences, we recalibrated the dataset 215 and calculated accuracy statistics for both the original (unadjusted) and adjusted RTS points (Table 1). 216 Two RTS surveys were conducted during helicopter flights in 2020 and 2023. We manually identified the exact locations of 217 aerial photos and created another RTS dataset. We then used it to perform an accuracy analysis in the central Gydan Peninsula 218 (Fig 9b, c). These points were also adjusted to our RTS mapping protocol, and the accuracy statistics were calculated for both 219 versions (Table 1). The performance of our dataset was evaluated using precision, recall, and F1-score, which integrates both 220 measures. In this context, precision refers specifically to the metric used in the F1-score calculation and should not be confused 221 with measurement precision, as no measurements were performed. Precision is calculated as the proportion of correctly 222 identified (true positive) RTS points when compared to the ground truth RTS points, among all mapped RTS points in the 223 dataset. Recall represents the proportion of correctly identified RTS points relative to the total number of RTS points in the 224 ground truth dataset. The F1-score is the harmonic mean of precision and recall, providing a balanced evaluation of both false

Figure 9 Field validation: (a) Locations of the Vaskiny Dachi research station with field survey area on Yamal Peninsula and helicopter survey area on central Gydan Peninsula, basemap: ESRI; (b) Photo of RTSs from the helicopter taken by Artem Khomutov, July 2023; (c) the same RTSs marked with the yellow point on the ESRI basemap, WorldView-2 24 July 2019. ESRI basemap used in (a) and (c) has the following credits: Esri, DigitalGlobe, GeoEye, i-cubed, USDA FSA, USGS, AEX, Getmapping, Aerogrid, IGN, IGP, swisstopo, and the GIS User Community.

Table 1 Number of RTSs used for the location accuracy analysis. The unadjusted number of RTSs represents the initial amount of RTSs in the ground truth datasets. The adjusted number of RTSs represents the amount of RTSs in the ground truth datasets adapted to the RTS mapping protocol applied for manual collection.

| Vaskiny Dachi Research Station<br>Survey 2024 |          | Gydan Helicopter Survey 2020 |          | Gydan Helicopter Survey 2023 |          |
|-----------------------------------------------|----------|------------------------------|----------|------------------------------|----------|
| unadjusted                                    | adjusted | unadjusted                   | adjusted | unadjusted                   | adjusted |
| 158                                           | 132      | 60                           | 39       | 12                           | 12       |

## 2.3.2 Classification accuracy

To assess the subjectivity of the classification, we conducted an experiment in which five co-authors of this study were tasked with classifying a subsample of 120 randomly stratified RTS points that equally covered all three types of morphology. The decision-tree schemes and the collection of screenshots of different RTSs were used as supportive materials (see Supplement).

We calculated the proportion of the same classifications by 5 co-authors compared to the original dataset and Jensen-Shannon distances explaining the deviation of classifications.

## 3. Results

#### 3.1 RTS points

The dataset is presented in a GeoPackage vector file of point geometry with 6168 RTS point locations. Mapped RTSs were distributed unevenly, covering Tazovsky Peninsula where no RTS were found, the Yamal Peninsula except its northern part, and covering the Gydan Peninsula except its southern part (Fig.10a). RTSs were significantly clustered according to Ripley's K function on a wide range of distances (p-value=0.001). The majority of areas of both peninsulas had less than 20 RTSs per 30\*30 km hexagon grid cell, indicating distinct hotspots of RTS occurrence with more than 100 RTSs per grid cell. The main areas with high RTS density were the western part of central Yamal and the area between the southern-western and north-eastern parts of central Gydan. On Gydan, they clustered along a distinct linear feature on its southern edge, south of which RTSs abruptly become almost absent (Fig.10b).

Figure 10 Distribution of all mapped RTSs: (a) Manually mapped RTSs (purple dots); (b) Density map of RTSs per  $30 \times 30$  km hexagonal grid cell. Projection: WGS 84 UTM Zone 43. Basemap: OSM Standard.

# 3.2 Terrain position

More than 75% of all RTSs were found at lakeshores (Fig.11a). The high-density areas of lakeshore RTSs correspond to RTS occurrence hotspots in the western part of central Yamal and the area between the south-western and north-eastern parts of central Gydan (Fig.11c).

The density of RTSs at the sea coasts was mostly less than 10 RTSs per grid cell. The highest density of coastal RTSs was found along the northern shores of Yuribei Bay in south-western Yamal (Fig.11b). For RTSs along river banks, gullies, and slopes, the predominating values of density were less than 10 RTSs per grid cell, not showing any spatial clustering (Appendix A).

Figure 11 Distribution of all mapped RTSs: (a) Manually mapped RTSs classified by location; Density maps of RTSs per  $30 \times 30$  km hexagonal grid cell located along the (b) seacoast and (c) lakeshores. Projection: WGS 84 UTM Zone 43. Basemap: OSM Standard.

# 3.3 Morphology

The majority (72%) of RTSs were classified as thermocirques, one-quarter of all RTSs are combined landforms, and less than 3% were classified as thermoterraces (Fig.12a). The majority of RTSs in all categories have a spatial density of less than 15 RTSs per grid cell.

Figure 12 Distribution of all mapped RTSs: (a) Manually mapped RTSs classified by morphology; Density maps of RTSs per  $30 \times 30$  km hexagonal grid cell classified as (b) thermocirque, (c) thermoterrace, and (d) a combination of both. Projection: WGS 84 UTM Zone 43. Basemap: OSM Standard.

Thermocirques were highly concentrated in hotspot areas of general RTS abundance (Fig. 10b). Combination landforms followed the high RTS abundance pattern mostly in the Gydan Peninsula but less so on the Yamal Peninsula. In contrast, thermoterraces lacked distinct high-density hotspots.

## 3.4 Spatial organization

More than half of all RTSs (64%) were classified as complex landforms and slightly more than one-third (36%) as single landforms (Fig. 13). Both complex and single landforms followed the general spatial distribution patterns, with high-density areas being located in the western part of the central Yamal Peninsula and the southern-western and north-eastern parts of the central Gydan Peninsula. The most frequent density range for both classes was less than 10 RTSs per grid cell.

Figure 13 Distribution of all mapped RTSs: (a) Manually mapped RTSs classified by spatial organization; Density maps of RTS per  $30 \times 30$  km hexagonal grid cell classified by spatial organization as (b) single or (c) complex landforms. Projection: WGS 84 UTM Zone 43. Basemap: OSM Standard.

#### 3.5 Concurrent processes

More than half (53.8%) of all RTSs were found to have at least one concurrent process detected, more than a third (33.4%) of all RTSs showed only one process detected, while much fewer RTSs demonstrated two or more processes detected at the same time (Fig. 14a). Lateral thermo-erosion and thermokarst were two very abundant RTS-concurrent processes (Fig. 14b). For the

cases where only one process was detected per RTS, there was a predominance of thermokarst (38%) followed by lateral thermo-erosion processes (30%) (Fig. 14c).

Using chord diagrams (Fig. 14d, e, f) allowed a depiction of the co-occurrence of concurrent processes estimated for the cases when two, three, or four processes were detected for RTS. In general, the co-occurrence of the concurrent processes shows different results depending on the cases of the amount of the processes detected. There was a clear trend of the co-occurrence of nivation and lateral thermo-erosion among all 3 cases (Fig. 14d, e, f). The co-occurrence of lateral thermo-erosion and ice-wedge erosion gradually increased with more processes detected. The co-occurrence of the nivation and the coastal thermo-erosion, when only 2 processes are detected, was relatively low but increased significantly with more processes detected. The presence of thermokarst processes, in general, decreased with more processes detected.

Figure 14 Results of concurrent processes detected for each RTS: (a) Pie-chart of the number of concurrent processes detected for each RTS; (b) Histogram representing the total count of all concurrent processes identified in mapped RTS; (c) Histogram representing the distribution of concurrent processes when only 1 process per RTS was detected. Chord diagrams representing the occurrence of concurrent processes in the case when (d) two concurrent processes were detected, (e) three concurrent processes were detected, and (f) four concurrent processes were detected. The size of the outer frame corresponds to the count of each concurrent process. The lines connecting color-coded concurrent processes stand for the co-occurrence: the thicker the line, the higher the co-occurrence.

Figure 15 Density maps of RTSs per  $30 \times 30$  km hexagonal grid cell classified by the presence of concurrent process: (a) Lateral thermo-erosion; (b) Coastal thermo-erosion; (c) Ice wedge erosion; (d) Nivation; (e) Thermokarst subsidence. Projection: WGS 84 UTM Zone 43. Basemap: OSM Standard.

RTSs attributed with concurrent processes exhibit low densities, with fewer than 5 RTSs per grid cell, regardless of the type of concurrent process (Fig. 15). RTSs with lateral thermo-erosion detected had higher densities in the western part of the central Yamal Peninsula and the central and northern Gydan Peninsula, with a hotspot in central Gydan Peninsula (Fig. 15a). RTSs with concurrent coastal thermo-erosion had higher densities in the western part of central Yamal and the north-western Gydan peninsulas, with three hotspots located at south-western part of central Yamal Peninsula, and northern and north-western Gydan Peninsula. (Fig. 15b). In general, the spatial distribution of RTSs with coastal thermo-erosion did not follow the main spatial patterns detected in the Fig. 10b. RTSs with ice wedge erosion had higher densities on the northern Gydan Peninsula and rather lower densities on the Yamal Peninsula, with one hotspot located on central Yamal Peninsula (Fig. 15c).

The spatial distribution of RTSs with concurrent ice wedge erosion also did not follow the main spatial patterns detected in Fig. 10b. RTSs with nivation had higher densities in central and northern Gydan Peninsula (more than 30 RTSs per grid cell) and rather lower (less than 15 RTSs per grid cell) densities on Yamal Peninsula (Fig. 15d). There were four hotspots: one on central Yamal Peninsula and three on central Gydan Peninsula. The spatial distribution of RTSs with nivation also did not follow the main spatial patterns detected in Fig. 10b. RTSs with concurrent thermokarst did follow the main spatial patterns detected in Fig. 10b and thus had higher densities and some hotspots in the western part of central Yamal Peninsula and the area between the southern-western and north-eastern parts of central Gydan Peninsula (Fig. 15e).

#### 4. Discussion

# 4.1 Data limitations

The manual collection of RTS points using the ESRI satellite basemap was effective across a large region but also had several limitations. One challenge was the resolution and zoom limitations, as the minimum detectable landform width was 20 meters, potentially excluding smaller features. Seasonal variability of the images in the ESRI satellite basemap further complicated the process, with snowpacks identifiable only in summer images, excluding all autumn (September) imagery. On the other hand, more extensive snow cover on certain images obscured some areas, hindering the accurate inventory of RTS and their attributes in these regions. Additionally, visual artifacts (blur, glare, clouds, contrails) in some imagery led to the omission of some cells, though this accounted for less than 0.5% of the total dataset. Temporal constraints posed another issue, as working with a single satellite image captured at a specific time could mean that some features were not visible or detectable under those conditions, leading to potential underrepresentation of RTS features. The rapid evolution of RTS in this area (i.e., 35% increase in RTS number in the central Yamal key site over 8 years reported by Ardelean et al., 2020) added difficulty for static inventory not only in the amount, with two RTSs of a single morphology potentially merging into a complex morphology, creating challenges in morphology classification. Similar challenges were reported in the literature (Huang et al., 2020; Rodenhizer et al., 2024). Additionally, updates to the ESRI satellite basemap during the mapping effort sometimes introduced inconsistencies across different stages of our workflow, e.g. between the initial mapping of RTS as points, the subsequent addition of attributes, and the later correction loop (Fig. 2). To alleviate some of these challenges, we effectively used the ESRI Wayback time series to verify uncertain landforms or attributes. Visual identification also had several challenges. Stabilized RTSs were difficult to recognize. Challenges were also faced when classifying partially stabilized RTS. The limitations concerning distinguishing slowly stabilizing slumps from stabilized slumps using optical data were also reported in the literature (Bernhard et al., 2020). The sediment accumulation as a secondary indicator for coastal thermo-erosion was found to be debatable due to its temporary nature. Some landforms, such as curved riverbanks, wave-cut lakeshores, active layer detachments (ALDs), and first-stage thermokarst mound (baydzherakh) development, could have been easily misclassified as RTS, leading to false positives in the final dataset.

Atypical for this area, Yedoma RTSs found in our inventory in the northern Gydan region differed significantly in appearance from the majority of the rest mapped RTSs. Yedoma deposits in West Siberia were not included in the Circum-Arctic Map of the Yedoma Permafrost Domain (Strauss et al., 2021), yet were described in the fieldwork in northern Yamal coast and northern Gydan coast (Vasil'chuk and Vasil'chuk, 2018; Vasilchuk et al., 2022). Since Yedoma mapping was not the aim of this inventory, we did not mark Yedoma RTSs. Moreover, Yedoma RTS's visual characteristics were not properly addressed in the initial visual identification protocol, leading to potential misidentifications.

#### 4.2 Accuracy

dataset uncertainties. For RTS mapping, this has been demonstrated before in a mapping exercise with multiple operators with varying degrees of experience (Nitze et al., 2024). Our subjectivity assessment using a subset of 120 RTS samples revealed that 3–16.6% were classified as non-RTS, with an average false positive rate of approximately 8.5% and a median of 4.1%. Consequently, the accuracy of our dataset based on this experiment averages around 0.91. We acknowledge that involving additional experts in visual correction could have improved accuracy and reduced subjectivity.

Human subjectivity, even if mapping is conducted by experienced researchers, can influence the results and contribute to

- The degree of classification similarity among the five co-authors, compared to the original dataset, exhibited a clear trend influenced by spatial organization, morphology, and two concurrent processes—coastal thermo-erosion and lateral thermoerosion—which were generally the most subjective. Spatial organization emerged as the most subjective parameter, with classifications showing the alignment in only half of the 120 sample points on average (Fig. 16a).
- To further quantify classification variability, we calculated Jensen-Shannon distances (Fig. 16b), a metric for measuring similarity between probability distributions. This value ranges from 0.0, indicating identical distributions, to 1.0, representing completely distinct distributions. The results confirmed the overall trend of morphology, coastal thermo-erosion, and lateral 376 thermo-erosion being the most subjective parameters, except for spatial organization, which showed minor differences in probability distributions. Coastal thermo-erosion exhibited the highest variation in classification probability distributions, likely due to two distinct hotspots observed in the heatmap.
- 379 Overall, the probability distributions of most classified parameters were either highly or moderately similar to those in the 380 original dataset. This suggests a generally consistent perception of RTS classification among the co-authors in the experiment. 381 RTS location accuracy was estimated for the area around the Vaskiny Dachi research station in central Yamal and central 382 Gydan Peninsulas, with helicopter surveys conducted in 2020 and 2023 (see Appendix B). RTS location accuracy assessments 383 for all areas revealed very high precision compared to the ground truth, confirming the reliability of the dataset (Table 2). A 384 relatively low recall, even after applying mapping style adjustments, indicates an approximate 50% underestimation of small 385 RTSs in the study area (Table 2) primarily due to the reasons described in the Data Limitations section (see Sect. 4.1). Please, 386 note that in this context, precision specifically refers to the metric used in the F1-score calculation and should not be mistaken 387 for measurement precision, as no actual measurements were conducted.

Figure 16 Classification subjectivity assessment: (a) Heatmap of the proportion of similar classifications by five co-authors compared to the classification in the dataset.; (b) Heatmap of Jensen-Shannon distances explaining deviation of classifications by five co-authors compared to the classification in the dataset.

Table 2 Average results of RTS location accuracy assessment for all three sets of ground truth field data: central Yamal and central Gydan (2020 and 2023). The adjusted value represents the accuracy measure calculated by comparing our dataset to the ground truth datasets adapted to the RTS mapping protocol applied for manual collection. The unadjusted value represents the accuracy measure calculated by comparing our dataset to the original ground truth datasets.

| Average results | for all three sets | of ground truth | field data |
|-----------------|--------------------|-----------------|------------|
|                 |                    |                 |            |

|           | Adjusted to the mapping style | Unadjusted to the mapping style |
|-----------|-------------------------------|---------------------------------|
| Precision | 0.96                          | 0.96                            |
| Recall    | 0.44                          | 0.38                            |
| F1 score  | 0.60                          | 0.54                            |

The relatively low F1 scores observed in our study can be attributed primarily to high underestimation (i.e., low recall) when compared to field data. Manual mapping of RTS using remote sensing data is often regarded as the most accurate approach (Swanson and Nolan, 2018; Segal et al., 2016a, b; Young et al., 2022; Luo et al., 2022). Efforts to enhance accuracy, particularly in terms of precision, have been made by incorporating multi-year datasets (Huang et al., 2021) and conducting multiple rounds of expert review (Segal et al., 2016b; Young et al., 2022). To ensure the reliability of manual mapping, Young et al. (2022) employed aerial field survey data for visual validation; however, their study did not report the initial recall of manual RTS mapping against field observations.

To the best of our knowledge, there are no existing studies that quantitatively assess the recall uncertainty of RTS manual mapping using remote sensing compared to field data, particularly over large spatial extents. Lewkowicz and Way (2019) attempted to estimate recall accuracy for manual RTS mapping in Banks Island, Canada (70000 km²), but their evaluation was based on a comparison with another remote sensing dataset rather than ground-based field observations. This limitation is largely due to the challenges associated with field data collection in remote study areas. Moreover, since field data provides only a single snapshot in time, some RTS classified as false positives based on remote sensing data may be true RTS that were simply not captured in the field dataset.

Despite these uncertainties, manually mapped RTS datasets serve as validation sources for automated deep-learning-based mapping algorithms (Nitze et al., 2021; Yang et al., 2023; Xia et al., 2022; Huang et al., 2021). Notably, relatively high F1 scores (F1  $> \sim 0.7$ ) for automated RTS mapping have been reported, but these assessments were primarily conducted against internal training datasets covering limited spatial extents and derived from manual mapping rather than field data (Huang et al., 2020; Nitze et al., 2021; Witharana et al., 2022; Yang et al., 2023).

- Our findings demonstrate that manual mapping using remote sensing data cannot be considered a definitive ground truth and
- is associated with a certain degree of inaccuracy, particularly concerning recall.
- Our accuracy assessment highlights the overall subjectivity in defining RTS morphology and spatial organization. These
- parameters critically influence what is visually identified as RTS in satellite imagery. This subjectivity aligns with previous
- RTS mapping experiments, where "mapping style" and the scientific background of domain experts were found to impact RTS
- delineation (Nitze et al., 2024). Our results demonstrate that, despite standardized instructions, both morphology and spatial
- organization remain the most subjective parameters in RTS classification.

# 4.3 Data applicability

- The collected data on RTSs holds significant potential for future applications and research across various disciplines. It can
- serve as a foundation for a more detailed characterization of the permafrost region. The spatial distribution and clustering of
- RTSs in West Siberia, combined with cryostratigraphic and geomorphological analyses, can help unravel driving processes
- and improve our understanding of these dynamic landforms.
- This dataset can also guide further research efforts, such as field surveys aimed at monitoring cryogenic processes as well as
- studies to uncover the ground ice origin. In addition, it provides a valuable reference for ground-truthing in machine learning
- applications, enabling more accurate automated remote sensing classifications and predictive modeling.
- The dataset is particularly relevant to ecologists, biogeochemists, geomorphologists, climatologists, permafrost scientists,
- hazard researchers, and remote sensing specialists. This data can also be useful in the context of managing permafrost-related
- risks and planning sustainable development in vulnerable regions.

### 435 **5. Data availability**

The dataset is available at Nesterova et al., 2025 (https://doi.pangaea.de/10.1594/PANGAEA.974406).

## 6. Conclusions

- In this study, we present the first large-scale manual RTS mapping effort with accuracy assessments based on field data. We
- present a comprehensive, manually mapped dataset of 6168 current retrogressive thaw slumps (RTS) for a large region in the
- West Siberian Arctic. Each RTS in the dataset was classified according to its morphology, spatial organization, terrain position,
- and concurrent permafrost relief-forming processes. Accuracy assessments with independent field data and expert knowledge
- indicate a high accuracy of the dataset while also highlighting some subjectivity in the classifications. Due to resolution
- limitations in the satellite image basemaps used for mapping, the dataset may underestimate the occurrence of small RTS in
- the region, resulting in an overall conservative estimate. Despite these constraints, our new RTS inventory offers valuable

insights for a wide range of research fields aiming at further investigations of RTS formation and dynamics, permafrost-climate interactions, permafrost-ecosystem feedbacks, and ground ice distribution in West Siberia.

# Appendices

# Appendix A

Figure A1. Density maps of RTS points counted per  $30 \times 30$  km hexagonal grid cell located at the (a)river, (b) gully, and (c) slope. Projection: WGS 84 UTM Zone 43. Basemap: OSM Standard.

# Appendix B

Table B Results of RTS location accuracy assessment for all three sets of ground truth field data: central Yamal Vaskiny Dachi research station and central Gydan Helicopter Survey (2020 and 2023). The adjusted value represents the accuracy measure calculated by comparing our dataset to the ground truth datasets adapted to the RTS mapping protocol applied for manual collection. The unadjusted value represents the accuracy measure calculated by comparing our dataset to the original ground truth datasets.

| Vaskiny Dachi research station, central Yamal |                               |                                 |  |  |
|-----------------------------------------------|-------------------------------|---------------------------------|--|--|
|                                               | Adjusted to the mapping style | Unadjusted to the mapping style |  |  |
| Precision                                     | 0.88                          | 0.88                            |  |  |
| Recall                                        | 0.44                          | 0.37                            |  |  |
| F-1 score                                     | 0.59                          | 0.52                            |  |  |
| Gydan Helicopter Survey 2020                  |                               |                                 |  |  |
|                                               | Adjusted to the mapping style | Unadjusted to the mapping style |  |  |
| Precision                                     | 1                             | 1                               |  |  |
| Recall                                        | 0.3                           | 0.2                             |  |  |
| F-1 score                                     | 0.46                          | 0.33                            |  |  |
| Gydan Helicopter Survey 2023                  |                               |                                 |  |  |
| Precision                                     |                               | 1                               |  |  |
| Recall                                        |                               | 0.58                            |  |  |
| F-1 score                                     |                               | 0.73                            |  |  |

## **Author contribution**

- NN: conceptualization, resources (data collection and correction), investigation, and writing (original draft preparation). IT:
- conceptualization, resources (data collection and correction), and writing (review and editing). ML: conceptualization,
- supervision, and writing (review and editing). AKh: resources (field data collection) and writing (review and editing). AK:
- writing (review and editing). IN: writing (review and editing). GG: supervision, and writing (review and editing).

# 475 **Supplement**

470

480

- The supplement of the decision-tree schemes and the collection of screenshots of different RTSs to help classify RTSs in West
- Siberia is available online at Zenodo (Nesterova and Tarasevich, 2025, https://doi.org/10.5281/zenodo.15063753).

# 478 Competing interests

The authors declare that they have no conflict of interest.

## Acknowledgments

- NN was funded by a DAAD fellowship ("STIBET-I"). IT, ML, and AKh were funded by the state assignment of the Ministry
- of Science and Higher Education of the Russian Federation (grant no. FWRZ-2021-0012). IN and GG were funded by the
- German Federal Ministry for Economic Affairs and Climate Action (BMWK) project ML4EARTH, European Space Agency
- (ESA) CCI+ Permafrost, as well as NSF (NSF Opp: #1927872 and #2052107) and google.org "Permafrost Discovery
- Gateway". AK was funded by the Lomonosov Moscow State University state assignment "The cryosphere evolution under
- climate change and anthropogenic impact" (#121051100164-0).

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
