# Peer review of "High-resolution inventory and classification of retrogressive thaw"

_Earth System Science Data, 2025_

## Author Comment (AC1)

**Author Response to Reviewer #1.**

*The comments by Reviewer #1 are in black. The author's responses are in blue. The changes suggested to the revised manuscript are in green in bold.*

*Anonymous Referee #1*

*Referee comment on "High-resolution inventory and classification of retrogressive thaw slumps in West Siberia" by Nina Nesterova et al Earth Syst. Sci. Data Discuss. [preprint], https://doi.org/10.5194/essd-2025-164, 2025.*

Review #1

The submitted manuscript, "High-resolution inventory and classification of retrogressive thaw slumps in West Siberia," presents an extensive update to the existing RTS inventory by manually mapping over 6,000 features across a vast region (445,226 km²) using multi-source, high-resolution satellite basemaps from 2016 to 2023. The study aims to enhance our understanding of RTS distribution, scale, and environmental controls in the West Siberian Arctic, a region where abrupt permafrost thaw remains poorly characterized.

This is a highly valuable and timely contribution to the field of permafrost research. The study is methodologically robust, clearly structured, and well contextualized within the broader scope of Arctic environmental change. The manual mapping approach, supported by field knowledge and verification, lends high confidence to the dataset, which achieves >90% accuracy in RTS identification. The comprehensive classification scheme—incorporating morphology, terrain position, and associated processes—greatly enhances the utility of the dataset for geomorphological, ecological, and climate-related applications.

Moreover, the dataset's potential as a reference for future remote sensing and machine learning studies is significant. The clarity of the methods, the transparent discussion of limitations, and the open-access nature of the data all reflect strong scientific standards and ensure broad usability. Overall, this is an excellent and much-needed piece of work, and I strongly support its publication in Earth System Science Data.

We would like to thank the reviewer for finding the time to review our manuscript. We highly appreciate valuable comments that help to improve the quality of the manuscript.

I have only a few minor observations that may help further improve the manuscript:

- You mention that a preliminary inventory of RTSs was published by Nesterova et al. (2021), but that it likely underestimated RTS distribution due to limited resolution. However, it is not entirely clear whether the RTSs previously mapped in that study were integrated into the current inventory, or if this work represents a completely new and independent mapping effort. If this is a fully new mapping process, it would be helpful to explicitly reference the earlier inventory in the Discussion section and highlight how the present approach offers improved results, particularly in terms of accuracy, resolution, or spatial coverage.

We have adjusted the text to make it clear that the previous RTS dataset is integrated. Lines 80-82 in the revised manuscript (Introduction):

We provide an extensive update of the West Siberian RTS inventory for 2021, which was performed by manually mapping RTS in the north of West Siberia using multi-source and multi-year satellite basemaps (high-resolution ESRI, Google Earth, and Yandex Maps satellite basemaps). We further added all the RTS locations reported for this region in the literature so far.

Lines 98-106 in the revised manuscript (Methodology):

2.1 RTS point mapping

The study area in the north of West Siberia is 445 226 km2 and includes the Yamal, Gydan, and Tazovsky peninsulas (Fig. 3). To ensure the completeness of the RTS dataset we reviewed previously published RTS datasets for the region, all of which were mapped using automated methods (Fig. 3). We manually filtered RTS datasets from Nitze et al. (2018), Runge et al. (2022), Bernhard et al. (2022), Huang et al. (2023), and Nitze et al. (2024b) to verify the presence of RTS and ensure that only true positives were included. This verification was conducted using the same available datasets that we later used for manual point collection, as described further below.

**Manually collected RTS dataset published in 2021 (Nesterova et al., 2021) was also integrated: points were revised, classified, and renamed.**

- One notable limitation of the study is that the RTS inventory was compiled using satellite imagery from various years. While it appears that the dataset spans approximately a decade, the exact temporal coverage is not clearly stated. For clarity and transparency, it would be helpful to synthesize all data sources and acquisition years in a concise table. Recent studies have shown that RTS activity in regions like the Yamal Peninsula is highly dynamic. For example, as you mention (line 56), new RTSs have emerged in recent years, and Ardelean et al. (2020) reported that in a small area of the Yamal Peninsula, the number of RTSs increased from 24 to 37 between 2004 and 2012. This raises the question of how reliable and representative a static inventory can be, considering that the number of RTSs may increase by 20% or more after a single warm season in some areas.

While I appreciate that this limitation is acknowledged in the Discussion, it may also be helpful to provide readers with a clearer temporal context for the mapped RTSs. For instance, including the year of observation for each RTS in the database, and possibly summarizing this information in a simple figure (perhaps overlaid in Fig. 4b), could give users a better sense of when most RTSs were identified. This would also strengthen the interpretation of the dataset's temporal relevance and help users better understand its limitations and potential applications.

We fully agree that the decade-long span of the dataset is a clear limitation that is inherited from the source. Unfortunately, it reflects the ESRI World Imagery mosaic, which aggregates scenes from multiple years. Please note that the acquisition year of the ESRI basemap used for each mapped RTS is recorded in the feature-level metadata of the inventory. To clarify that, we added it to the text. Lines 128-134 of the revised manuscript (Methodology):

The majority of the high-resolution satellite images used in the ESRI basemap mosaic are recent Maxar images obtained after 2015 (Fig. 4b). Over a third of the study area is covered by satellite images from 2023 (Fig. 4b). Since the ESRI basemap was utilized as the primary source, all metadata related to the satellite images **(image date acquisition, image resolution, image accuracy, min and max map level, satellite description, ESRI release name)** in the mosaic for identifying the RTS is stored within the inventory dataset's metadata. Yandex Maps basemap presents a mosaic of various satellite imagery taken in 2016-2018 with spatial resolutions ranging from 0.4 up to 15 m. The majority of images are dated July 2017 (Nesterova et al., 2021). For the Google satellite image layer, no individual image metadata was provided.

We added some text to describe the limitations of static inventory that are addressed in the Discussion. Lines 341-346 of the revised manuscript (Discussion):

Temporal constraints posed another issue, as working with a single satellite image captured at a specific time could mean that some features were not visible or detectable under those conditions, leading to potential underrepresentation of RTS features. **The rapid evolution of RTS in this area (i.e., 35% increase in RTS number in the central Yamal key site over 8 years reported by Ardelean et al., 2020) added difficulty for static inventory not only in the amount**, with two RTSs of a single morphology potentially merging into a complex morphology, creating challenges in morphology classification. Similar challenges were reported in the literature (Huang et al., 2020; Rodenhizer et al., 2024).

We fully agree that non-static dynamic data can reveal further RTS characteristics. This dataset aims to provide an overview of RTS locations and spatial distribution using available high-resolution imagery. We focus on the dynamics of RTSs in the next research.

- Have you observed any particular patterns in the distribution of RTSs in relation to yedoma deposits? The manuscript does not make it clear whether yedoma terrains are more or less favorable to RTS development. While I know that this region may not be particularly representative for extensive yedoma deposits, it could still be helpful to briefly address this aspect in the Discussion. Even a short statement noting the presence or absence of RTS in yedoma areas—or the limitations of assessing this due to their restricted extent—would add value and context for readers interested in the geomorphic controls on RTS formation.

Thank you for pointing this out! We added some context based on the scientific literature available regarding the topic of Yedoma in West Siberia. The lines 357-363 in the revised manuscript (Discussion):

**Atypical for this area, Yedoma RTSs found in our inventory in the northern Gydan region differed significantly in appearance from the majority of the rest mapped RTSs. However, Yedoma deposits in West Siberia were not included in the Circum-Arctic Map of the Yedoma Permafrost Domain (Strauss et al., 2021), yet were described in the fieldwork in northern Yamal coast and northern Gydan coast (Vasil'chuk and Vasil'chuk, 2018; Vasilchuk et al., 2022). Since Yedoma mapping was not the aim of this inventory, we did not mark Yedoma RTSs. Moreover, Yedoma RTS's visual characteristics were not properly addressed in the initial visual identification protocol, leading to potential misidentifications.**

- Lines 44-45, please add a citation.

Thank you, we added. Lines 44-46 of the revised manuscript (Introduction):

Active RTS can be considered as one of the clear indicators of permafrost response to increased air temperatures and higher summer precipitation (**Lantz and Kokelj, 2008; Kokelj et al., 2015; Leibman et al., 2021; Barth et al., 2025**).

- Line 48: I couldn`t find Jones et al., 2019 at the Bibliography. I recommend also to refer to the study by Barth et al. (2023).

Thank you, that was a typo, we corrected it to "Ward Jones et al., 2019", and we added the new article of Barth et al., 2025. Lines 44-46 of the revised manuscript (Introduction):

Active RTS can be considered as one of the clear indicators of permafrost response to increased air temperatures and higher summer precipitation (**Lantz and Kokelj, 2008; Kokelj et al., 2015; Leibman et al., 2021; Barth et al., 2025**).

- Line 50: Are you sure that all the study area is in continuous permafrost? According to Obu et al (2019) model the southern part of peninsulas are in discontinuous permafrost.

Thank you, we corrected the text. Lines 51-54 of the revised manuscript (Introduction):

**The north of West Siberian Arctic, with its predominantly continuous permafrost distribution (Obu et al., 2019), is characterized by a high abundance of RTS.** The prevalence of massive ground ice (Baulin et al., 1967; Streletskaya et al., 2013; Leibman and Kizyakov, 2007; Badu, 2015) that often occurs close to the surface contributes to the widespread abundance of RTSs (Khomutov et al., 2017).

- Line 93: would be good to list the names of peninsula on the map (Figure 3). In addition, would be good to overlap the type of permafrost on Fig. 3 (you can yo use Obu et al., 2019), or at least the limit between continuous and discontinuous permafrost.

Thanks, we improved Figure 3 by 1) adding the Permafrost occurrence map by Obu et al. (2019) as a basemap; 2) by adding the names of the Peninsulas.

- Line 187: coastal is correct here?

Yes, we mean coastal thermo-erosion. This term is explained in lines 182-184 of the revised manuscript (Methodology):

The Coastal thermo-erosion classifier includes not only the sea coast erosion but also river and lakeshore erosion. It was determined by a sharp dark outline of the RTS base along the coastline of a waterbody and the absence of sediment accumulation in the water (Fig. 8a).

- Lines 237-238: these tests should be presented in the Methodology section.

Thank you, we moved it to the Methodology section.

- 13b: in the center of the Gydan Peninsula there is a hexagon with a 20-30 RTS per cell. Can you check if it is correct and there are coastal RTS there?

Figure 13b depicts Single RTSs regardless of their location. If you meant Figure 11b and the red hexagonal cell with 20-30 RTSs, then yes, it is correct; it is a very narrow part of Gyda Bay that is covered by hexagons.

- In the Introduction you start by saying that RTS are formed due to the thaw of exposed ice-rich permafrost. However, along the study there are no references to the warming of the climate in this area in the last decade or to a temperature increase of permafrost in the region. Would be useful to put the inventory in the climatic context since this phenomenon is climatically controlled and refer to climate evolution in Western Siberia.

Thank you. We added some clarifications to the text of the Introduction. The lines 51-56 of the revised manuscript (Introduction):

**The north of West Siberian Arctic, with its predominantly continuous permafrost distribution (Obu et al., 2019), is characterized by a high abundance of RTS.** The prevalence of massive ground ice (Baulin et al., 1967; Streletskaya et al., 2013; Leibman and Kizyakov, 2007; Badu, 2015) that often occurs close to the surface contributes to the widespread abundance of RTSs (Khomutov et al., 2017). **Moreover, the observed amplification of seasonal thawing and growth of permafrost temperatures (Babkina et al., 2019; Biskaborn et al., 2019; Vasiliev et al., 2020) presents an additional factor for the mass initiation of RTS in the region.**

References:

Ardelean et al., 2020. doi:10.3390/rs12233999

Barth et al., 2023. https://doi.org/10.1594/PANGAEA.961794

Obu et al., 2019. https://doi.org/10.1016/j.earscirev.2019.04.023

All references added. Thank you again for your time!
* * *
**Author Response to Reviewer #2.**

*The comments by Reviewer #2 are in black. The author's responses are in blue. The changes suggested to the revised manuscript are in green in bold.*

*Anonymous Referee #2*

*Referee comment on "High-resolution inventory and classification of retrogressive thaw slumps in West Siberia" by Nina Nesterova et al Earth Syst. Sci. Data Discuss. [preprint], https://doi.org/10.5194/essd-2025-164, 2025.*

Review #2

The submitted manuscript, "High-resolution inventory and classification of retrogressive thaw slumps in West Siberia," presents a new and comprehensive inventory of 6,168 retrogressive thaw slumps (RTSs) in the West Siberian Arctic, using high-resolution imagery. The work is scientifically valuable and fills a geographic gap in permafrost monitoring efforts, offering one of the first large-scale, high-resolution inventories from this understudied region. It fills a substantial geographic gap in RTS inventories, especially in a region where abrupt thaw processes are underdocumented.

The classification scheme and the spatial scale elevate the value of this dataset. The overall structure is clear and transparent, and the open-access release will allow broad scientific reuse. Nevertheless, some clarifications and minor improvements would enhance the dataset's interpretability, reproducibility, and scientific context.

We are grateful to the reviewer for taking the time to evaluate our manuscript and for the constructive comments that have helped us improve it!

A few minor comments that may help further improve the manuscript:

- Line 84: The inventory provides point data for each RTS, but it is not explained whether this point corresponds to the slump's headwall or geometric center. I would suggest clarifying how point locations were assigned. A consistent rule, such as placing the point at the headwall or initiation zone, would be most relevant for geomorphological or modeling applications.

Thank you for pointing this out. We did our best to aim at the visual center of the landform. We clarified that in the text. The lines 135-139 of the revised manuscript (Methodology):

RTSs were identified at a 1:1000 mapping scale in the satellite imagery based on visual indicators such as a clear outline of the headwall, the presence of a mudflow, and the sharp contrast in colors between the disturbed slump floor with bare ground and the adjacent intact tundra vegetation (Fig.4c). Thus, stabilized RTSs were also identified when the indicators were still visible. **For each identified feature, we created a point in the location of the RTS within the visible outlines of the RTS with the best possible approximation to the visual center of the landform.**

- Line 103: The dataset uses 3.9 × 3.9 km grid cells for summarizing RTS densities.

  Is there a reason this specific resolution was chosen? Does it align with other pan-Arctic products, or was it selected to balance detail and generalization?

Yes, this cell size was found optimal for visual inspection and progress tracking. We added the statement about the balance to the text. The lines 111-114 of the revised manuscript (Methodology):

For our visual identification and manual collection of RTS points, we created a regular grid of 3.9 * 3.9 km cells covering the entire study area (Fig. 4a). **This cell size was chosen as the optimal for visual inspection of the area and progress tracking, balancing detail and generalization.**

- Line 134, you mentioned that the RTS points underwent two visual corrections by the first author. While this allows for consistency, it introduces potential subjectivity in feature recognition and classification thresholds. You mention subjectivity in the limitations, but I wonder if a second opinion would have increased the accuracy. Please acknowledge this as a limitation and suggest the value of inter-observer validation or consensus mapping.

Thank you for this comment. We believe that the rounds of correction should have increased the accuracy due to the prior experience of the first author in West Siberian RTS mapping using Yandex.Maps (Nesterova et al., 2021). In the previous inventory, the most ambiguous cases were discussed collaboratively among all co-authors. While we fully agree that the robustness of the dataset will increase with more independent verifications performed by co-authors, fully manually re-checking 6168 RTSs would not be feasible due to time constraints. We addressed this issue in the revised manuscript, lines 364-370 (Discussion):

4.2 Accuracy

Human subjectivity, even if mapping is conducted by experienced researchers, can influence the results and contribute to dataset uncertainties. For RTS mapping, this has been demonstrated before in a mapping exercise with multiple operators with varying degrees of experience (Nitze et al., 2024). Our subjectivity assessment using a subset of 120 RTS samples revealed that 3–16.6% were classified as non-RTS, with an average false positive rate of approximately 8.5% and a median of 4.1%. Consequently, the accuracy of our dataset based on this experiment averages around 0.91. **We acknowledge that involving additional experts in visual correction could have improved accuracy and reduced subjectivity.**

- Line 209: F1 scores vary substantially across validation subsets. While this is valuable transparency, the implications are not clearly discussed. Please add 2–3 sentences interpreting these metrics. For example, what factors drove the lowest scores (e.g., coarser imagery, seasonal effects, older basemaps)? Would you recommend restricting the use of the inventory for automated training to higher-confidence regions?

We discuss F1 score interpretation as well as address possible factors lowering the scores in the 4.2 Accuracy section. Lines 400-413 of the revised manuscript (Discussion):

The relatively low F1 scores observed in our study can be attributed primarily to high underestimation (i.e., low recall) when compared to field data. Manual mapping of RTS using

remote sensing data is often regarded as the most accurate approach (Swanson and Nolan, 2018; Segal et al., 2016a, b; Young et al., 2022; Luo et al., 2022). Efforts to enhance accuracy, particularly in terms of precision, have been made by incorporating multi-year datasets (Huang et al., 2021) and conducting multiple rounds of expert review (Segal et al., 2016b; Young et al., 2022). To ensure the reliability of manual mapping, Young et al. (2022) employed aerial field survey data for visual validation; however, their study did not report the initial recall of manual RTS mapping against field observations.

To the best of our knowledge, there are no existing studies that quantitatively assess the recall uncertainty of RTS manual mapping using remote sensing compared to field data, particularly over large spatial extents. Lewkowicz and Way (2019) attempted to estimate recall accuracy for manual RTS mapping in Banks Island, Canada (70000 km2), but their evaluation was based on a comparison with another remote sensing dataset rather than ground-based field observations. This limitation is largely due to the challenges associated with field data collection in remote study areas. Moreover, since field data provides only a single snapshot in time, some RTS classified as false positives based on remote sensing data may be true RTS that were simply not captured in the field dataset.

- Lines 340-345: The manuscript states that some mapped features were likely misclassified RTSs. Are these RTS flagged, removed, or included in the final dataset?

Thank you for pointing out the need for clarification in the text. All the misclassified RTSs revealed during the first and second rounds of the first author's correction were removed. In the text, we aimed to list possible unknown misclassifications that could have been missed. We clarified that we refer to the final dataset, lines 351-356 in the revised manuscript (Discussion):

Visual identification also had several challenges. Stabilized RTSs were difficult to recognize. Challenges were also faced when classifying partially stabilized RTS. The limitations concerning distinguishing slowly stabilizing slumps from stabilized slumps using optical data were also reported in the literature (Bernhard et al., 2020). The sediment accumulation as a secondary indicator for coastal thermo-erosion was found to be debatable due to its temporary nature. **Some landforms, such as curved riverbanks, wave-cut lakeshores, active layer detachments (ALDs), and first-stage thermokarst mound (baydzherakh) development, could have been easily misclassified as RTS, leading to false positives in the final dataset.**

- Lines 346-347: While the manuscript refers to Yedoma, there is no discussion of whether RTSs coincide with yedoma terrain, known to be highly susceptible due to excess ground ice. Maybe add a figure including yedoma distribution and the mapped RTSs.

Thank you for noticing the need for some background information on Yedoma in West Siberia. Unfortunately, there is no available map of Yedoma distribution in West Siberia to be added. The existing map by Strauss et al. (2021) does not cover West Siberia. We added some context based on the scientific literature available regarding the topic of Yedoma in West Siberia. The lines 357-363 in the revised manuscript (Discussion):

**Atypical for this area, Yedoma RTSs found in our inventory in the northern Gydan region differed significantly in appearance from the majority of the rest mapped RTSs. However, Yedoma deposits in West Siberia were not included in the Circum-Arctic Map of the Yedoma Permafrost Domain (Strauss et al., 2021), yet were described in the**

**fieldwork in northern Yamal coast and northern Gydan coast (Vasil'chuk and Vasil'chuk, 2018; Vasilchuk et al., 2022). Since Yedoma mapping was not the aim of this inventory, we did not mark Yedoma RTSs. Moreover, Yedoma RTS's visual characteristics were not properly addressed in the initial visual identification protocol, leading to potential misidentifications.**

- Finally, to better contextualize the presented dataset within the growing body of RTS studies, the authors may wish to cite recent advances in RTS.

Ardelean et al., 2020, **DOI**: https://doi.org/10.3390/rs12233999

Thank you, added. Lines 341-346 of the revised manuscript (Discussion):

Temporal constraints posed another issue, as working with a single satellite image captured at a specific time could mean that some features were not visible or detectable under those conditions, leading to potential underrepresentation of RTS features. **The rapid evolution of RTS in this area (i.e., 35% increase in RTS number in the central Yamal key site over 8 years reported by Ardelean et al., 2020) added difficulty for static inventory not only in the amount**, with two RTSs of a single morphology potentially merging into a complex morphology, creating challenges in morphology classification. Similar challenges were reported in the literature (Huang et al., 2020; Rodenhizer et al., 2024).

Makopoulou et al., 2024, **DOI**: https://doi.org/10.1002/esp.5890

Thank you, added. Lines 46-50 of the revised manuscript (Introduction):

RTS occurrence significantly impacts the environment by altering the vegetation, topography, hydrology, as well as carbon fluxes (Lantz et al., 2009; Thienpoint et al., 2013; Cassidy et al., 2017). The prediction of RTS occurrence and activity is challenging due to heterogeneous ground ice distribution (Pollard and French, 1980; **Makopoulou et al., 2024**) across the Arctic, limited observational field data (Ward Jones et al., 2019), and the lack of models capable of simulating RTS initiation and dynamics (Yang et al., 2025).

Yang et al., 2025 (ARTS Dataset), **DOI**: https://doi.org/10.1038/s41597-025-04372-7

Thank you, added. Lines 46-50 of the revised manuscript (Introduction):

RTS occurrence significantly impacts the environment by altering the vegetation, topography, hydrology, as well as carbon fluxes (Lantz et al., 2009; Thienpoint et al., 2013; Cassidy et al., 2017). The prediction of RTS occurrence and activity is challenging due to heterogeneous ground ice distribution (Pollard and French, 1980; Makopoulou et al., 2024) across the Arctic, limited observational field data (Ward Jones et al., 2019), and the lack of models capable of simulating RTS initiation and dynamics (**Yang et al., 2025**).

---

## Author Response (AR2)

**Author Response to Reviewer #2.**

The comments by Reviewer #2 are in black. The author's responses are in blue. The changes suggested to the revised manuscript are in green in bold.

Anonymous Referee #2

Referee comment on "High-resolution inventory and classification of retrogressive thaw slumps in West Siberia" by Nina Nesterova et al Earth Syst. Sci. Data Discuss. [preprint], https://doi.org/10.5194/essd-2025-164, 2025.

**Second Review #2**

I am pleased with the revisions made to the manuscript, which have further improved its clarity and quality. I thank the authors for carefully considering the comments and suggestions provided in the previous round. The manuscript is now scientifically sound and ready for publication after some minor modifications.

We would like to thank the reviewer for finding the time for the second review!

- Figure 1: The white line, which seems to represent the boundary of the study area, is not explicitly explained in the legend. Please clarify its meaning, and also adjust the placement of the scale bar text, as it currently overlaps with the white line and reduces readability.

Thank you for pointing this out. We have adjusted the map: moved features, so that they do not overlap, added study area to the legend. Moreover, we updated a reference from the dataset (Nitze et al., 2024) to the recently published data paper (Nitze et al., 2025). We also updated a list of references since one was found missed.

- Figure 14: The chord diagrams are very well presented; however, I did not find any information in the Methodology section about the software or package used to create them. Please add this detail for clarity and reproducibility.

Thank you for noticing. The chord diagrams were indeed plotted not in Python. We plotted it in R and added this information to the text:

**2 Methodology**

Our approach includes four main steps: (1) visual identification of RTS and manual RTS point collection, (2) classification and parameter attribution, (3) iterative correction loop, and (4) final accuracy assessment (Fig. 2). Manual RTS point collection, classification, and correction were performed in QGIS software version 3.14. Accuracy analysis, plotting, and statistical calculations were performed using Python version 3.12.7. **Chord diagrams were plotted in R, using RStudio 2024.12.0+467.** The resulting points were analysed for clustering using Ripley's K function. Ripley's K function determines whether spatial points have a random, dispersed, or cluster distribution over a certain distance or scale (Dixon, 2001).

- One category of processes responsible for the occurrence of retrogressive thaw slumps is nivation. As this term can be somewhat ambiguous, I suggest that the authors clarify it by explicitly referring to the contribution of persistent snow cover.

**We clarified that in the text:**

The influence of concurrent (happening in parallel to RTS development) processes on RTS development is described in Nesterova et al. (2024). For each mapped RTS, we noted the possible presence of 5 concurrent processes: lateral thermo-erosion, coastal thermo-erosion, ice wedge erosion, nivation, and thermokarst subsidence. Lateral thermo-erosion was identified by the rugged outline of the RTS and visible traces of erosive channels (Fig. 8a). The Coastal thermo-erosion classifier includes not only the sea coast erosion but also river and lakeshore erosion. It was determined by a sharp dark outline of the RTS base along the coastline of a waterbody and the absence of sediment accumulation in the water (Fig. 8a). We have noted ice wedge erosion when an RTS headwall had a jagged outline resembling the adjacent polygonal surface of undisturbed tundra (Fig. 8b). **Nivation in the context of this study is considered as persistent snow cover.** It was detected as white patches of snowpacks that stayed over the summer within RTS (Fig. 8a). Thermokarst subsidence appears as small thermokarst ponds filled with water. It is noticeable as black patches within the RTS outline (Fig. 8b).

| Thank you again for | or the reviews! |      |  |
|---------------------|-----------------|------|--|
|                     |                 |
 |  |